# Multi-modal omics analysis of a paediatric melanoma highlights mechanisms underlying treatment resistance

Marlena Mucha [1,2], Sebastian Bühner [1,2], Maurice Loßner[1,2], Victoria E. Fincke[1,2], Nic G. Reitsam [2,3], Felix Dorn[1,2], Dajana Lobbes[1,2], Katharina Gastberger [1,2], Tobias Schuster[4], Sebastian Dintner [2,3], Christopher Schroeder [5], Ines B. Brecht[6], Dominik T. Schneider [7], Bruno Märkl [2,3], Michael C. Frühwald [1,2] & Pascal D. Johann [1,2] ✉

## Abstract

**Background** Cutaneous malignant melanoma is a common cancer in adults but extremely rare in young children, affecting fewer than one child per million each year in Europe. Because of its rarity, most treatments for children are adapted from adult therapies, despite possible biological differences. This study aimed to explore the molecular features of a rare and aggressive melanoma in a 16-month-old patient to understand disease progression and treatment resistance.
**Methods** We studied the tumour and metastases of a patient with a melanoma carrying an *NRAS* mutation, who received chemotherapy and immune checkpoint inhibitor treatment. The patient died 10 months after diagnosis. We used DNA methylation analysis, single-nucleus RNA sequencing, and deep spatial transcriptomic profiling to examine genetic changes, gene activity, and their spatial distribution in both the primary tumour and lymph node metastases.
**Results** Here, we show that the tumour displayed high genetic and transcriptomic diversity. We identified increases in *MITF* and *BRAF* gene copies as likely key drivers of the aggressive disease, which were not detected at diagnosis. We also found activation of biological pathways, including VEGFA and WNT signalling, and abnormal activity of several genes linked to immune therapy response, with marked variation between tumour regions.
**Conclusions** This case demonstrates that paediatric melanoma can harbour complex and spatially variable molecular changes that contribute to rapid disease progression and treatment failure. Our findings support incorporating detailed spatial transcriptional profiling into clinical assessment to better guide therapy in rare paediatric cancers.

## Plain language summary

Melanoma is a type of skin cancer that is common in adults but extremely rare in very young children. Because it is so uncommon, children are usually treated with approaches designed for adults, which may not work as well. In this case study, we investigated an aggressive case of melanoma in a 16-month-old child to understand why the cancer progressed quickly and did not respond to treatment. We examined samples from the original tumour and its spread to the lymph nodes using advanced techniques to map genetic changes and patterns of gene activity. We found specific changes in key cancer-related genes and signals that likely drove the disease and made treatment ineffective. These results highlight the importance of detailed tumour analysis to guide better treatment strategies for rare childhood cancers.

Paediatric cutaneous melanoma is a rare malignancy, accounting for less than 1% of all melanoma cases diagnosed in Germany[1]. Due to its rarity as well as atypical clinical and histological features, it is often diagnosed late, with patients presenting with advanced disease. It can manifest as one out of three histopathological subtypes with varying clinical implications: conventional melanoma, bearing the closest similarity to adult UV-induced melanoma; typically less aggressive spitzoid melanoma; and melanoma arising from congenital nevi (CNM)[2]. Treatment options are often extrapolated from those applied to adult patients and include surgical resection and immune checkpoint

[1]Pediatrics and Adolescent Medicine, Swabian Children's Cancer Center, University Hospital Augsburg, Augsburg, Germany. [2]Bavarian Cancer Research Center (BZKF) and KIONET, Augsburg, Germany. [3]Pathology, Faculty of Medicine, University of Augsburg, Augsburg, Germany. [4]Pediatric Surgery Clinic, University Hospital Augsburg, Augsburg, Germany. [5]Institute of Medical Genetics and Applied Genomics, University Hospital Tübingen, Tübingen, Germany. [6]Department of Paediatric Haematology and Oncology, University Children's Hospital Tübingen, Tübingen, Germany. [7]Clinic of Pediatrics, Municipal Hospital Dortmund, University Witten/Herdecke, Witten, Germany. ✉e-mail: pascal.johann@uk-augsburg.de

inhibitor (ICI) therapy, and therapies targeting tumours harbouring specific mutations.

Recent advances in molecular profiling of tumours have greatly increased our understanding of adult melanomagenesis. Importantly, the Mitogen Activated Protein Kinase (MAPK) pathway has been identified as a major culprit, with *BRAF* V600, *NRAS* Q61 and *NF1* mutations driving 60%, 20% and 4% of melanomas, respectively[3,4]. These discoveries led to the development of targeted therapies, including Trametinib + Dabrafenib, approved for treatment of *BRAF*[V600E] mutated melanoma, and the combination therapy of Trametinib + Mebendazole for *BRAF*[WT]/*NRAS*[Q61K] melanoma[5,6]. Additionally, recent advancements in single-cell and spatial transcriptomics have provided useful insights into tumour heterogeneity, metastasis and development mechanisms for many cancer types, including melanoma[7–10].

In this study, we investigated a case of a unique paediatric metastatic melanoma patient with exceptionally rapid disease progression. Through extensive molecular analyses of the primary and metastatic lesions, we aimed at shedding light onto the potential mechanisms of disease development contributing to the aggressive phenotype, which could inform the optimal treatment approach for this patient in retrospect. Using DNA methylation profiling, we discovered somatic *MITF* and *BRAF* amplifications as the plausible 'second hits', and the drivers of the aggressive disease, in addition to the previously identified *NRAS* Q61H. Moreover, we observe upregulation of the WNT pathway signalling in metastases compared to the primary tumour. Using deep spatial transcriptomics and single-nucleus RNA sequencing (snRNA-Seq), we identified the possible section of the primary tumour and cell population, respectively, which likely contributed to metastases development, and implicated *VEGFA* as a spatially informed marker. Finally, we suggest mechanisms of resistance to ICI therapy in this patient.

Importantly, our report contributes to the very limited literature and molecular datasets on paediatric melanoma and illustrates the application of comprehensive molecular profiling. Additionally, we discuss clinical implications for diagnosis, prognosis, and therapeutic decision-making in rare paediatric malignancies such as melanoma.

## Methods

### Patient and tissue specimens
The patient was enroled in the STEP registry (Ethics Committee of the Friedrich-Alexander-Universität Erlangen-Nürnberg Reg. No. 4340), which aims to improve the epidemiological and clinical recording of children and adolescents with particularly rare tumours in Germany. Written informed consent has been obtained from the patient's parents for the use of the patient's material for research purposes. The FFPE blocks and fresh frozen material were obtained from the Biobank facility at the University Hospital Augsburg.

### Sanger sequencing
DNA from the fresh-frozen primary tumour material was isolated using the Qiagen All Prep Mini kit, and from the FFPE material (healthy lymph node and lymph node metastasis samples) using the Qiagen FFPE All Prep kit, according to the manufacturer's protocol.

The *NRAS* (OMIM 164790) exon 3 region was amplified by Polymerase Chain Reaction (PCR) using specific primers (Forward: 5′-CACCCCCAGGATTCTTACAG-3′, Reverse: 5′-TCGCCTGTCCTCATGTATTG-3′). PCR reactions were carried out in a total volume of 15 µl. Sanger sequencing reactions were performed using the BigDye™ Terminator v3.1 Cycle Sequencing Kit (Applied Biosystems) and analyzed on an automated capillary sequencer (SeqStudio Genetic Analyzer, Thermo Fisher). The resulting sequences were evaluated with Geneious® software and aligned to the NRAS reference sequence (RefSeq: NM_002524). Variants were classified based on the guidelines provided by the American College of Medical Genetics and Genomics (ACMG).

### DNA methylation profiling
DNA methylation profiling was performed using the Illumina Infinium HD methylation assay as per the manufacturer's protocol, which included the QC—qPCR and DNA restoration step of the FFPE sample (not necessary for FF samples).

The copy number profile analysis was performed using the R package *conumee* 2.0[11], based on data pre-processed with the *minfi* package[12]. To achieve this, we utilised an annotation object containing information about the available CpG sites and the localisation of the genes under investigation. Additionally, a genomically stable reference object was used to conduct tangent normalisation. Following this, the genomic bins were processed, and circular binary segmentation was applied, resulting in a segmentation file. The results were subsequently visualised using the *conumee* 2.0 package[11].

### Digital spatial profiling (DSP)
FFPE blocks were sectioned using a rotating microtome at 5 µm thickness and placed on standard, positively charged microscopy slides (TOMA). Slides were prepared for DSP according to the manufacturer's protocol. In brief, sections were deparaffinized and subjected to heat-induced protein epitope retrieval (using retrieval buffer at pH 9, Thermo Fisher Scientific, #00-4956-58) for 15 min at 100 °C using a steam cooker. Retrieval of mRNA was performed using proteinase K (Thermo Fisher Scientific, #25530049) at 1 µg/ml for 15 min. Slides were then incubated with the human Whole Transcriptome Atlas (WTA) probe panel (Nanostring, #NA-GMX-RNA-NGS-HuWTA-4, Lot HWTA21003) at 37 °C in a hybridisation oven overnight. Sections were washed according to standard protocols, and direct immunofluorescence was performed with a Melanoma morphology marker panel (Nanostring, #121300311) composed of S100B/Pmel17 (Novus NBP2-54426, Novus NBP2-34638), CD45 (Novus NBP2-34528) and a nuclear counterstain (SYTO-13), according to the manufacturer's dilution recommendations. Namely, 220 ul of morphology marker solution per slide was prepared, containing 22 µl of each SYTO-13 stain, S100B antibody and CD45 antibody, and 187 ul Buffer W.

GeoMx DSP device run and library preparation were performed according to the manufacturer's protocol. ROIs were selected based on their localisation within the tissue, guided by the melanoma morphology markers staining. All collected ROIs were circular and of the same size (70,693.66 µm²), except for the two healthy skin ROIs (31,334.67 µm²). No segmentation of ROIs was performed. The resulting library was sequenced by Novogene using an Illumina sequencer with the specifications provided by Nanostring (paired-end reads at length 27, index length 8) with 5% PhiX spike-in using a NovaSeq X Plus 25B PE150.

Raw FASTQ files were then processed using the DSP Standalone Software, and the resulting files were imported into R (v.4.3) for analysis using the *NanostringNCTools 1.10.1*, *GeomxTools 3.6.2* and *Geomx-Workflows 1.8.0* packages. Segment QC was performed using the following parameters: minSegmentReads = 1000, percentTrimmed = 80, percentStitched = 80, percentAligned = 80, percentSaturation = 50, minNegativeCount = 4, maxNTCCount = 1000, minNuclei = 100, minArea = 5000. All 39 collected ROIs passed QC. Genes detected in less than 10% ROIs were filtered out, resulting in 11,752 genes remaining, and the data were then nomalised using Q3 normalisation.

Dimensionality reduction on the resulting matrix was performed using the *Umap* R package and visualised with *ggplot2*. The list of differentially expressed genes between ROIs belonging to the primary tumour (Fig. 4a) was calculated as being the top 0.5% by standard deviation (99.5th percentile).

Bubble plots of gene expression were generated using *ggplot2*, with *x* and *y* coordinates aesthetics parameters as the 'ROI coordinate X' and 'ROI coordinate Y' from the ROI annotation file, and bubble size as the expression of a given gene, with scale_size_continuous set to the bottom and upper limits of expression of the gene. The resulting bubble plots were then imported into Adobe Illustrator, overlayed on top of the corresponding tiff

images collected from DSP and adjusted slightly to overlay the location of the respective ROI exactly while keeping the bubble size unchanged.

Differential gene expression analysis between Metastasis/Recurrence and Primary tumour ROIs was performed on the Q3-normalised data with the *DESeq2 1.42.1* package, with significantly deregulated genes defined by abs(log$_2$FoldChange) > 1 & baseMean > 15 & padj <0.05[13]. Results were visualised using the *EnhancedVolcano 1.2* and *pheatmap 1.0.12* R packages.

### snRNA library preparation and sequencing

Nuclei suspension for the snRNA-Seq preparation of the primary tumour was obtained according to Slyper et al. protocol[14]. SnRNA-Seq library consisting of 5000 nuclei was prepared using the Chromium Single Cell 3' v3.1 kit (10X Genomics) according to the manufacturer's protocol. The library was sequenced on Illumina NovaSeq X Plus 25B PE150 by Novogene, assuming approximately 50,000 reads per cell.

### snRNA-Seq analysis

The FASTQ files were subsequently processed with the *Cell Ranger (v5)* software using the *count* pipeline with default parameters, and the resulting filtered matrix was imported into R version 4.3.1. The matrix was then processed into a Seurat object using the *Seurat 5.0* package. Cells were then filtered based using the following parameters: nFeature_RNA > 200 & nFeature_RNA < 3500 & percent.mt <10 & percent.antisense <3.5 & percent.ribosomal <8. Additionally, doublets were identified and removed with *scDblFinder 1.17.2*. These quality control steps resulted in the final matrix consisting of 4901 cells and 36,601 genes. Data was then normalised using the LogNormalize (scale.factor = 10000) function within *Seurat*, highly variable features were identified using the FindVariableFeatures function, and data was scaled using ScaleData. Dimensionality reduction was performed with RunUMAP (dims 1:20), and cell clusters were identified using FindCluster at 0.5 resolution. Cluster gene markers were then identified with FindAllMarkers with logfc.threshhold = 0.2.

Cell clusters were annotated based on their gene markers using reference databases within the *celldex 1.12* package, such as Blueprint Encode Data and Human Primary Cell Atlas Data, using the *SingleR 2.4.1* package.

Dot plots, bar plots and snRNA-Seq heatmaps were generated using the *SCPubr 2.0.2* package[15].

Copy number variations (CNVs) were inferred from snRNA-Seq using the *InferCNV 1.18.1* package with cutoff = 0.1, denoise = true and HMM = false, mapped onto human genome assembly hg38_gencode_v27, and visualised using the plot_cnv function within InferCNV.

Melanoma cell state modules were assigned to the malignant cells using the AddModuleScore function within Seurat, with top gene markers for each module as 'features'[16], and the top-scoring module was assigned to each cell.

T cell populations were projected onto a reference atlas of mouse tumour-infiltrating T cells according to[17] using the *ProjecTILs 1.0.0* package and visualised with plot.projection().

### Statistics and reproducibility

The manuscript includes data from multiple tissues collected from one patient. For the GeoMx DSP dataset, multiple biological replicates are represented by the numbers of ROIs belonging to the same tissue type (primary tumour, healthy lymph node, lymph node metastases, recurrence). No data were excluded from the analyses in our study.

## Results

### Case presentation

In this case study, we present a 16-month-old female patient diagnosed with congenital malignant melanoma. Already at birth, the patient had several naevi on the scalp. Prior to diagnosis, the patient had been seen at multiple clinics, and the naevi were then closely monitored, but the recommended surgery was postponed until clear development of lymph node metastases. In November 2016, she was diagnosed with melanoma in the right parietal region (3 cm × 1.3 cm × 3 cm) with no signs of infiltration or penetration of

the skullcap (Fig. 1a). Concurrently, she presented with several pathologically enlarged, partially cystic lymph nodes in the right cervical region (bottom arrow in the second picture in Fig. 1a). Histopathological and genetic analysis of this primary tumour revealed it to be *BRAF* wildtype, *NRAS* mutated (c.183 A > C, p. Q61H, COSM 586), GD2 and PD-L1 negative (<1% cells stained). The patient underwent multiple surgeries (Fig. 1b, in red), including resection of the primary tumour, lymph node metastases and parotidectomy, and recurrence. The wound from the primary surgery was stitched using the rotation flap technique. Following the final surgery, she received seven doses of Pembrolizumab, from which she experienced severe side effects. Additionally, she received a combination treatment of Trametinib and Mebendazole, as well as three courses of chemotherapy (Fig. 1b, in blue). Unfortunately, despite an initial good response to chemotherapy, the disease ultimately recurred. Following the last scan, less than 9 months after diagnosis (Fig. 1a, last picture) and identification of extensive metastases to the lungs and spine, the patient was placed in palliative care. She passed away five weeks later.

To profile the genetic and transcriptional landscapes of the primary tumour and metastases, we performed single-nucleus RNA sequencing (snRNA-Seq), spatial transcriptomics, DNA methylation array and Sanger sequencing. The tissue source (formalin-fixed paraffin-embedded (FFPE) or fresh frozen) used for each experiment is indicated in Fig. 2a.

### Somatic *MITF, BRAF* amplifications and NRAS Q61H as the disease drivers

Sanger sequencing on the primary tumour revealed an *NRAS* c.183 A > C, p.Q61H (COSM 586) mutation (Fig. 2b) and no *BRAF* mutations. Since at the time of diagnosis, only tumour DNA was profiled by Sanger sequencing, we first aimed to determine whether the *NRAS* mutation was somatic or germline. Therefore, we performed Sanger sequencing on DNA extracted from the primary tumour, healthy lymph node, and lymph node metastasis. This analysis confirmed that the patient did not have a germline *NRAS* mutation, and the previously identified c.183 A > C substitution was heterozygous (Fig. 2b). While *NRAS* Q61 activating mutations are common drivers of adult and paediatric melanoma, this specific residue substitution (Q61H) has recently been characterised as the least aggressive in driving melanoma development, compared to other Q61 missense mutations[18].

Therefore, we hypothesised that the tumour acquired additional aberrations which, together with the *NRAS* Q61H, would contribute to the aggressive disease progression.

We performed DNA methylation-based CNV analysis on the primary tumour, healthy lymph node and lymph node metastasis present at the time of initial diagnosis. This revealed no major copy number changes between the primary tumour and lymph node metastasis, confirming the latter as a true metastasis rather than a synchronous tumour (Fig. 2c). Interestingly, we noted an increase in copy number of the part of chromosome 3p encompassing the *MITF* gene locus only in the tumour tissues (Fig. 2d). While MITF (microphthalmia-associated transcription factor) is melanocyte-specific transcription factor, and as such is a known oncogene responsible for driving 15–20% of adult metastatic melanomas[19–21], it is rarely mutated in children. In fact, recent analysis of Italian paediatric melanoma patients revealed *MITF* mutations in only 3/123 of the cohort, all causing the p.E318K substitution[22]. Another study identified a *MITF* amplification in 1/23 profiled patients, to our knowledge the only *MITF*-amplified paediatric melanoma patient reported up to date[23]. Therefore, albeit rare, *MITF* amplifications can occur in paediatric melanoma and should be included in the genetic analysis of the tumour during diagnosis.

Additionally, we noted an amplification of the entire chromosome 7, including the *BRAF* locus (Fig. 2c, d). While this aberration is less locus-specific than the *MITF* amplification, BRAF is a well-described melanoma oncogene[3,24] and thus this amplification could have also contributed to the progression of the disease.

Based on these results, we hypothesise that the increased copy number of *MITF in concert with BRAF* was the secondary, acquired disease driver,

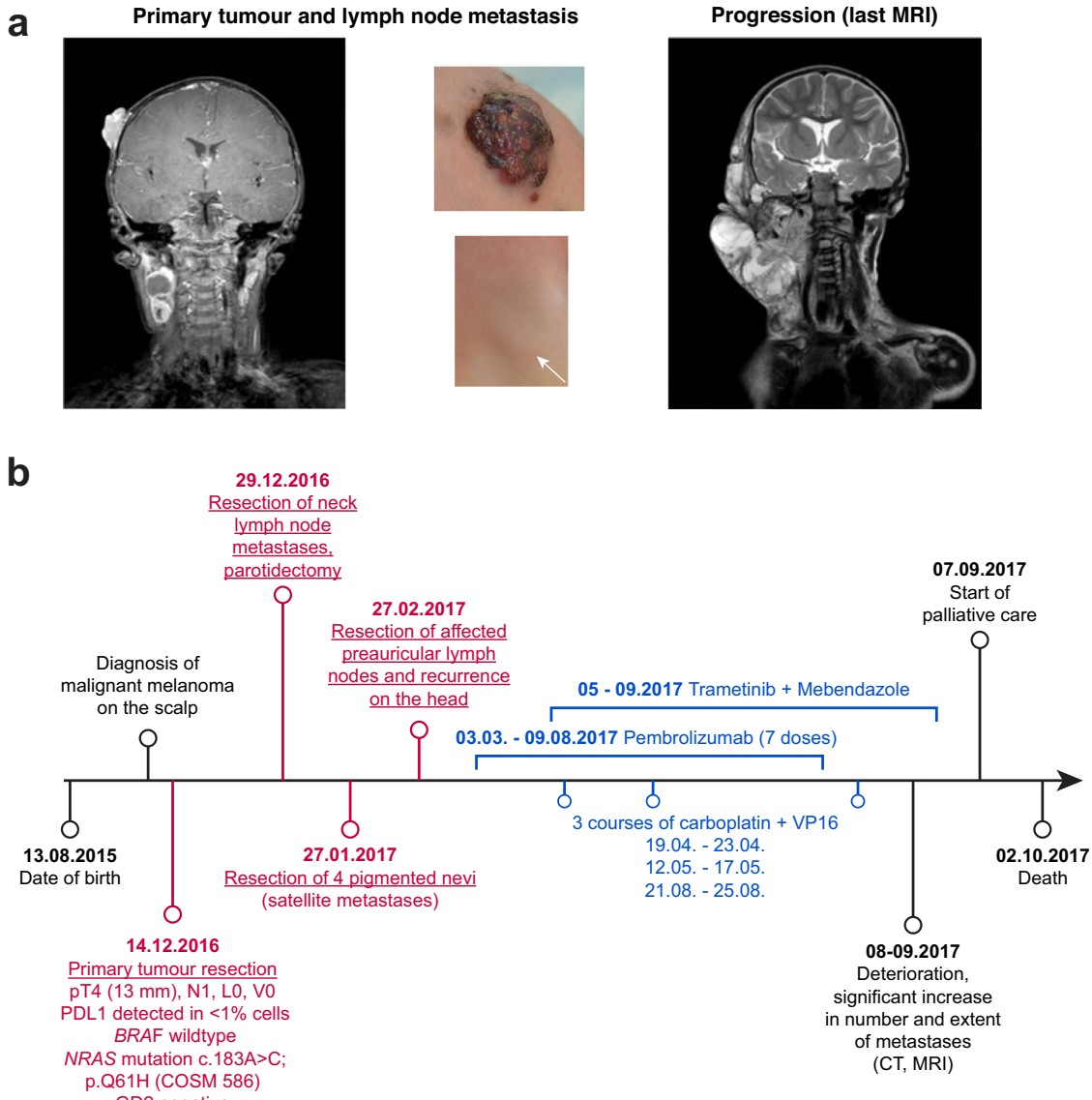

**Fig. 1 | Clinical presentation and treatment history. a** Images of the tumours at different stages throughout disease development: primary tumour and lymph node metastasis at diagnosis, and final progression prior to palliative care. **b** Timeline of the disease progression and therapy. Outlined in red are the surgeries, in blue therapeutic treatments.

which, together with the less penetrant *NRAS* Q61H, was the genetic cause of the aggressive course of the disease.

### *MITF* and *BRAF* are overexpressed in the tumour tissues

Next, we wished to determine whether the *MITF* and *BRAF* amplifications resulted in overexpression of the respective proteins and then to determine the transcriptional heterogeneity of the tissues. To enable this, we performed single-nucleus RNA sequencing (snRNA-Seq) of the fresh frozen primary tumour material (Fig. 2a, e), as well as deep spatial profiling using the GeoMx DSP platform.

We performed 3' snRNA-Seq using the 10X Genomics Chromium platform to gain an unbiased overview on cell populations contained in the primary tumour. Following the quality control steps to remove cells with low gene detection, high mitochondrial and ribosomal signatures, as well as cell doublets, we retained 4901 cells for downstream analysis. The data was then log normalised, scaled, and dimensionality reduction was performed prior to clustering. We detected 10 transcriptionally distinct cell clusters, as visualised by UMAP (Fig. 2e, upper panel). Using the top 20 gene markers, we annotated the clusters as those belonging to the tumour (0, 1, 2, 3, 4, 9) or

microenvironment (5, 6, 7, 8) (Fig. 2e). Then, we profiled the CNV landscape in the tumour clusters using the *InferCNV* R package, with the healthy cell clusters as reference (Fig. 2f). This mRNA-based CNV profile of the primary tumour closely resembled that obtained from DNA methylation data, including the presence of amplification of the chromosome bands encompassing *MITF* and *BRAF*.

Then, we measured the expression levels of *MITF*, *BRAF* and *NRAS* in the annotated cell types, which revealed increased expression of the two former genes in the tumour cells, compared to the microenvironment (Fig. 2g).

To orthogonally validate this finding as well as gain a more thorough picture of the heterogeneity of the tumour tissues, we employed deep spatial profiling using GeoMx DSP technology, which combines immunofluorescent imaging with whole transcriptome barcoding and subsequent profiling of self-defined tissue regions (Regions of Interests, ROIs). We prepared FFPE slides containing tissue samples from the primary tumour and neighbouring skin, lymph node metastases (including a resected healthy lymph node), and recurrence site (Supplementary Fig. 1a). To guide ROIs selection, we stained the tissues using antibodies targeting S100B/

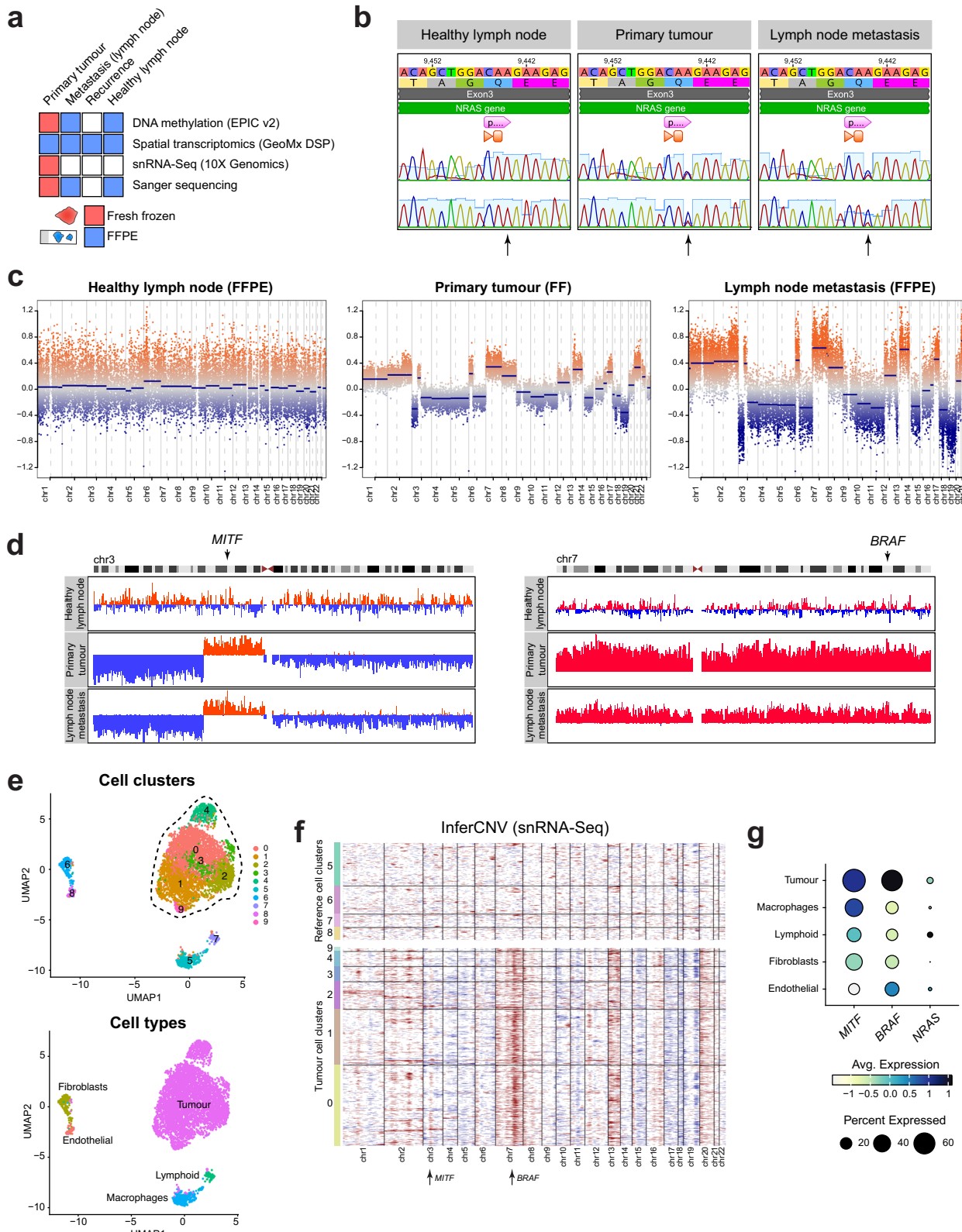

**Fig. 2 | Somatic NRAS Q61H mutation, and *MITF* and *BRAF* amplifications as the disease co-drivers. a** Schematic representation of the techniques used to analyse the fresh frozen and formalin-fixed paraffin embedded (FFPE) material. **b** Sanger sequencing of *NRAS* from genomic DNA extracted from the healthy lymph node and primary tumour. **c** Genome plots representing DNA methylation-derived copy number variation (CNV) profiles from healthy lymph node, primary tumour and lymph node metastasis, originating either from formalin-fixed paraffin-embedded (FFPE) or fresh frozen (FF) tissue samples. **d** IGV genome browser views of the DNA methylation-derived copy number variation (CNV) profiles from healthy lymph node, primary tumour and lymph

node metastasis, zoomed in to a 189 Mb region surrounding the *MITF* locus on chromosome 3 (left), or showing the entire chromosome 7 including *BRAF* (right). Marked in red are amplifications, in blue deletions. **e** UMAP visualisation of all single cells (*n* = 4901) within the primary tumour, coloured by clusters identified with *Seurat* (top panel), or by cell type annotation (bottom panel). **f** A heatmap showing snRNA-Seq derived copy number variations in all primary tumour cells, ordered by Seurat clusters as per Fig. 2e. Healthy cell clusters 5, 6, 7 and 8 were used as a reference. Indicated with arrows are loci containing *MITF* and *BRAF*. **g** A dot plot showing scaled expression of *MITF*, *BRAF* and *NRAS* genes in the snRNA-Seq cell types.

PMEL for melanoma cells, CD45 for immune cells and SYTO13 nucleic acid stain. Subsequently, the slides were incubated with a Whole Transcriptome Atlas from Nanostring (Bruker) comprising *circa* 18,000 probes targeting the human transcriptome and processed using the GeoMx DSP. In total, we profiled the transcriptome from 39 circular, non-segmented ROIs: $n = 22$ from the primary tumour (Supplementary Fig. 1a, Images 1 and 2), $n = 8$ from the lymph node metastasis present at diagnosis (Image 3), $n = 2$ from recurrence (Image 5), and $n = 7$ representing healthy tissues (ROIs marked with arrows).

Following the ROI quality control, gene filtering and normalisation, the resulting gene expression matrix used for downstream analysis included 11,752 genes. Visualisation of the gene expression patterns within each ROI in Uniform Manifold Approximation Projection (UMAP) showed strong similarity of ROIs belonging to the same tissues, with healthy tissues clustering closely together, as expected (Supplementary Fig. 1b). Confirming the snRNA-Seq results, *MITF* and *BRAF* showed higher expression levels in the ROIs located in tumour tissues, with *MITF* showing a bigger difference (Supplementary Fig. 1c).

### Transcriptional signatures leading to ICI resistance
Next, we focused on the transcriptional heterogeneity within the primary tumour and between the primary tumour and its metastases to pinpoint potential mechanisms responsible for the lack of response to immune checkpoint inhibition (ICI) therapy.

Within the GeoMx datasets, we notably observed strong heterogeneity between the ROIs from the primary tumour (Supplementary Fig. 1b). Particularly, ROIs B09-B12 displayed the closest similarity to the neighbouring healthy skin ROIs (B07, B08), while ROI B06 showed the closest similarity to the recurrence tissue (Supplementary Fig. 1b, also marked with a red square in Supplementary Fig. 1a).

First, we sought to determine the transcriptional evolution between the metastases compared to the primary tumour. Using *DESeq2*, we performed differential gene analysis of ROIs belonging to the lymph node metastases and recurrence, compared to the primary tumour ROIs (Fig. 3a, b)[13]. This revealed upregulation of the WNT signalling genes such as *WNT3* (LFC = 1.63, padj = 5.04e−35), *WNT10A* (LFC = 1.03, padj = 1.09e−15) and *WNT8A* (LFC = 1.77, padj = 6.51e−22). As the WNT pathway is a key signalling cascade implicated in carcinogenesis (reviewed extensively in ref. 25,26), we speculate that its overexpression contributed to the fast development of the metastases as well as relapse. Additionally, increased WNT/beta-catenin signalling has previously been implicated in resistance to immune therapy in other cancers, suggesting it as a potential reason for resistance in this patient as well[27].

Moreover, intrigued by the significantly increased expression of the *CXCL9* cytokine in recurrent tumour ROIs compared to primary (Fig. 3b), we wished to examine the expression levels of other cytokines. Non-hierarchical clustering revealed increased expression of macrophage migration inhibitory factor (*MIF*) in the tumour ROIs, highest in the lymph node metastases, suggesting involvement of inflammatory macrophages in metastasis development and ICI evasion.

Therefore, using GeoMx spatial profiling of tissue regions from the primary tumour, metastasis, and recurrence, we determined possible causes of metastasis and relapse resistance to ICI therapy with Pembrolizumab.

### Resistance to ICI could be explained by expression pattern of six previously implicated genes
Recently, a machine learning model for predicting response to ICI has been developed, based on RNA-Seq data from patients treated with PD-1/PD-L1 inhibitors across 18 solid tumours, including melanoma[28]. Based on this model, another recent study identified a set of 6 genes whose expression patters correlated with response to ICI in a metastatic melanoma patient cohort[29]. Specifically, expression of 4 genes (*CD163, SAMSN1, ITGAX* and *TNFAIP2*) positively correlated with progression-free survival and response to ICI treatment, while expression of 2 genes (*MTSS2* and *PSMB5*) correlated negatively. Given that our patient did not respond to ICI with

Pembrolizumab, we analysed the expression patterns of those six genes in our GeoMx data to provide a retrospective rationale for her lack of response and subsequent rapid disease progression. A comparative analysis of expression of these genes in tumour and healthy tissues ROIs revealed increased expression of *MTSS2* and *PSMB5*, and decreased expression of *CD163, SAMSN1, ITGAX* and *TNFAIP2* genes in most of the tumour ROIs, suggesting an upregulation of ICI-resistance-driving transcriptional programmes in the tumour and microenvironment (Fig. 3d, f). Similarly, *PSMB5* and *MTSS2* were expressed higher in the snRNA-Seq tumour cells (Fig. 3e). This result, together with the upregulation of WNT signalling (Fig. 3a, b) and *MIF* (Fig. 3c), could explain the patient's lack of response to ICI and rapid disease progression.

### *VEGFA* activation exhibits spatial variability within the primary tumour
The results presented in Figs. 3 and 4 suggested potential mechanisms behind the development of metastases and relapse, and their resistance to ICI. However, we were also interested in the heterogeneity of the primary tumour itself. Based on Supplementary Fig. 1b, we hypothesised that the peripheral regions of the primary tumour (ROIs B06 and B09-B012) were most likely to give rise to metastatic lesions, as they displayed stronger similarity to the non-primary tumour tissues. Furthermore, we wished to find genes which could serve as potential biomarkers for prognosis.

We ranked the GeoMx gene expression matrix based on standard deviation within primary tumour ROIs and plotted the expression of the top 0.5% most variable genes (Fig. 4a). We noted *VEGFA* as a gene displaying particularly heterogenous expression between ROIs (Figs. 3c and 4a, b). VEGFA (vascular endothelial growth factor A) plays a crucial role in melanoma progression through its involvement in angiogenesis, immune evasion, and metastasis promotion[30,31].

To ensure that it was indeed the melanoma cells expressing high levels of *VEGFA* and not endothelial cells of the blood vessels, we first evaluated the expression of *VEGFA* and an endothelial cell marker *PECAM1* across all ROIs. This revealed a lack of positive correlation of expression of the two genes, with the highest levels of *PECAM1* in ROIs belonging to healthy tissues - C08, C12 and D01-D03 (Supplementary Fig. 2a). Then, we investigated the high-resolution images obtained with GeoMx DSP for the presence of blood vessels in the A05-A07 and B02 ROIs. While we did not stain for PECAM1, those ROIs contain very little CD45 staining, while ROI B02 contained a high number of immune cells, confirming the lack of clear correlation. In those ROI,s we also do not see any blood vessel structures (Supplementary Fig. 2b).

We confirmed these findings in the snRNA-Seq dataset, where, as expected, *PECAM1* expression levels were the highest in the endothelial cell population, followed by macrophages, whereas *VEGFA* expression levels were highest in the macrophages and tumour cells (Supplementary Fig. 2c, d). Therefore, we believe that the *VEGFA* expression comes from melanoma cells.

Taken together, heterogeneous expression of *VEGFA* may result in uneven responses to therapeutic treatments, particularly anti-angiogenic or ICI, making a combination therapy potentially beneficial[32,33]. Additionally, areas with high *VEGFA* expression are more likely to contain cells which intravasate and seed metastases. Based on these results, we hypothesise that patients with variable expression of *VEGFA* might benefit from a combination therapy of VEGFA-targeted therapy, such as Bevacizumab, with ICI, such as Pembrolizumab.

### Transcriptional heterogeneity of the primary tumour
Next, we wished to investigate the cellular composition of the primary tumour in more detail. To this end, we utilised the snRNA-Seq dataset, from which we subset only the tumour cells identified by aberrant CNV profiles (Fig. 2f) and repeated the clustering method to identify distinct cell populations within the primary tumour only (resolution = 0.3). This identified five sub-clusters, characterised by expression of gene markers such as *EYA1* and *SOX6* (cluster 0), *ENO4* and *FBXL7* (cluster 1), *EEF1A1* and *TPT1*

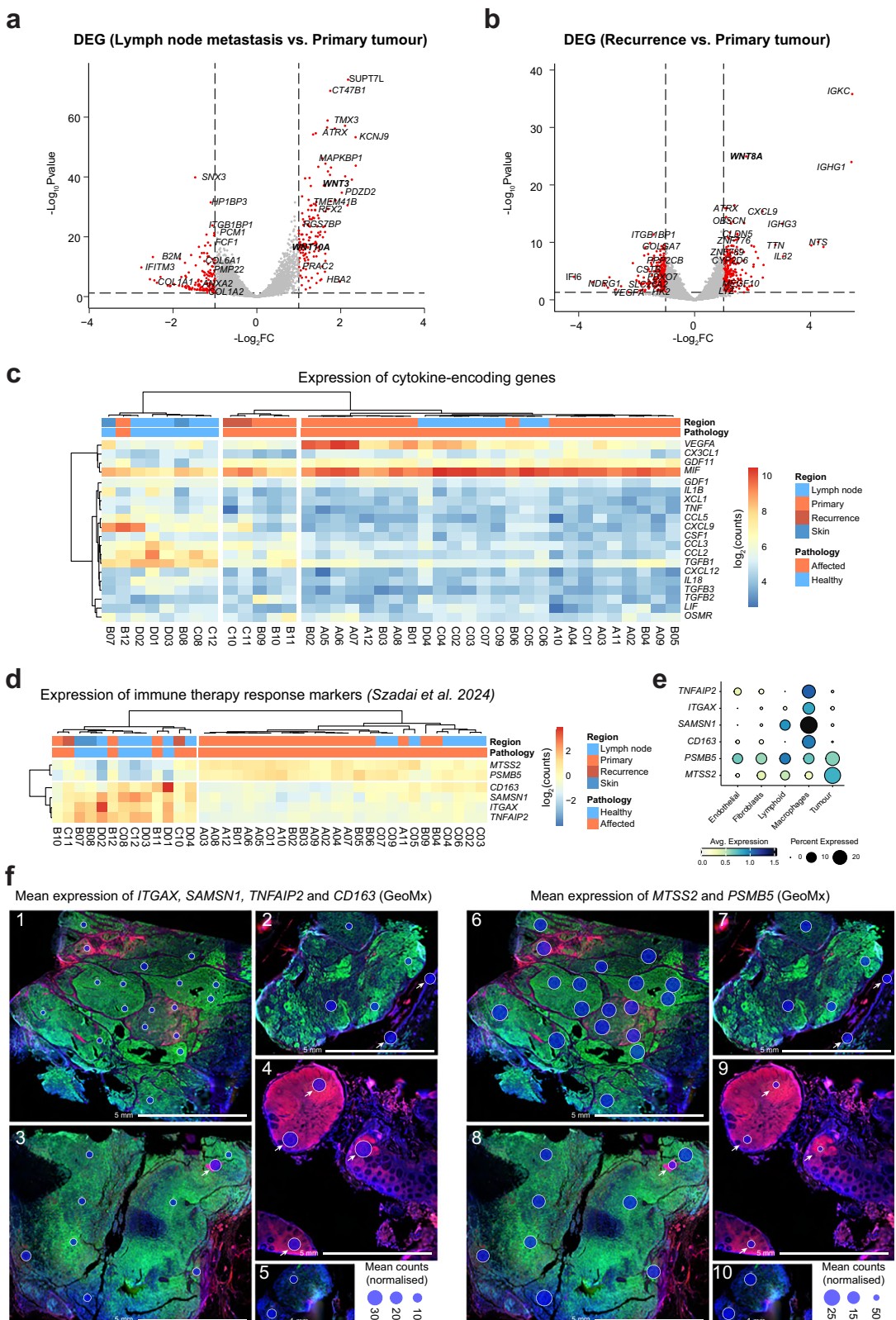

(cluster 2), *ASPM* and *DIAPH3* (cluster 3), *SEMA3* and *P3H2* (cluster 4) (Fig. 5a, b). Expression of these markers suggests highly proliferative, DNA-damage induced signatures, as expected from a malignant tumour. Importantly, all clusters expressed high levels of mitochondrial and DNA-damage related genes.

Then, aiming to assess whether any of the tumour clusters have shifted from a differentiated melanocytic state to a more undifferentiated or mesenchymal-like/invasive state[34], we compared the expression levels of melanocytic (*TYR, MLANA, DCT* and *MITF*) and mesenchymal (*ZEB1, VIM, FN1* and *AXL*) marker genes in the tumour clusters. This revealed

**Fig. 3 | Transcriptional signatures leading to ICI resistance. a** Volcano plot of differentially expressed genes between lymph node metastasis and primary tumour ROIs. The $\log_2$ fold change in expression ($\log_2$FC) is plotted against -$\log_{10}$pvalue. **b** Volcano plot of differentially expressed genes between recurrence and primary tumour ROIs. The $\log_2$ fold change in expression ($\log_2$FC) is plotted against $\log_{10}$pvalue. **c** Non-hierarchical clustering of expression ($\log_2$(normalised counts)) of cytokine-encoding genes derived from the GeoMx ROIs. **d** Non-hierarchical clustering of expression ($\log_2$(normalised counts)) of *MTSS2, PSMB5, CD163, SAMSN1, ITGAX* and *TNFAIP2* derived from the GeoMx ROIs. **e** Dot plot showing

normalised expression of *MTSS2, PSMB5, CD163, SAMSN1, ITGAX* and *TNFAIP2* in the snRNA-Seq cell types. **f** Left: Bubble plot of mean expression of the four genes (*ITGAX, SAMSN1, TNFAIP2* and *CD163*) whose higher expression in melanoma correlates positively with response to ICI, overlaid on top of DSP images. Right: Bubble plot of mean expression of the two genes (*MTSS2* and *PSMB5*) whose higher expression in melanoma correlates negatively with response to ICI, overlaid on top of DSP images. Bubble size represents mean counts (normalised). Arrows represent healthy tissue ROIs. Images 1, 2, 6, 7: primary tumour; Images 3, 8: lymph node metastases; Images 4, 9: healthy lymph node; Images 5, 10: recurrence.

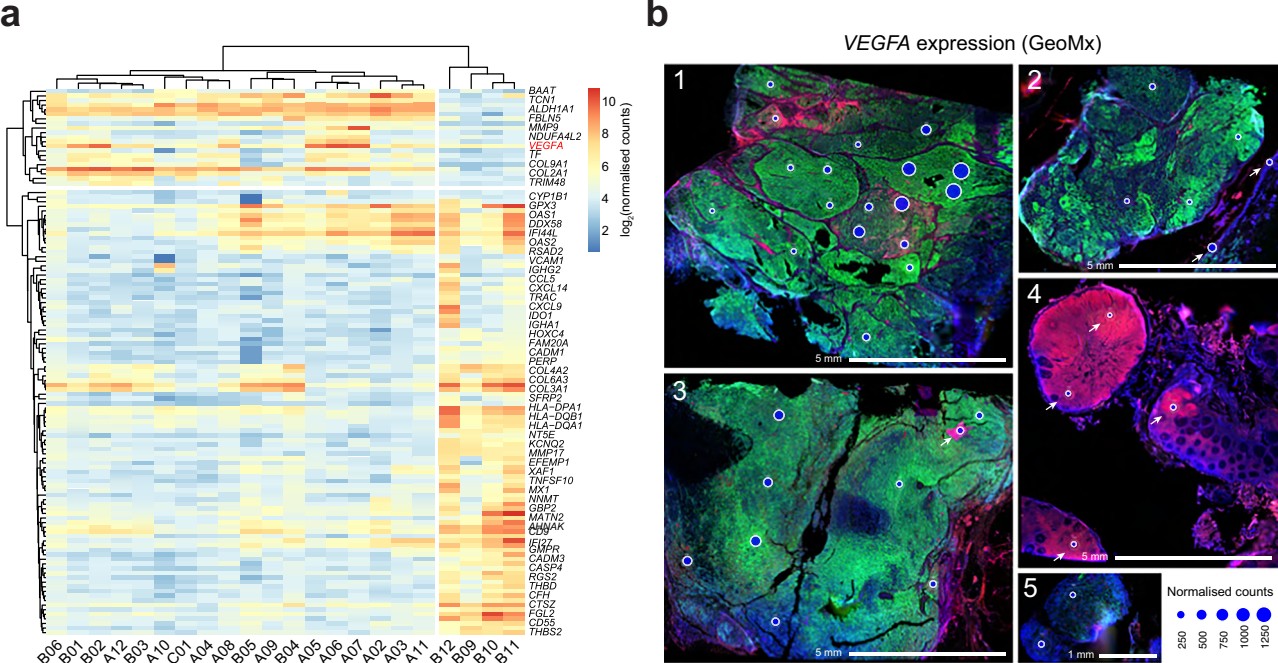

**Fig. 4 | *VEGFA* activation exhibits spatial variability within the primary tumour. a** Non-hierarchical clustering of the top 0.5% differentially expressed genes within the primary tumour ROIs, calculated by standard deviation of $\log_2$(normalised counts). **b** Bubble plot of *VEGFA* gene expression (normalised counts) represented as bubble size, overlaid on top of images from GeoMx DSP. Images are numbered as in Supplementary Fig. 1a.

potential dedifferentiation of cells within tumour cluster 4, which expressed lower levels of melanocytic markers compared to other clusters (Fig. 5c, d).

Additionally, we utilised a recently published set of malignant melanoma transcriptional states and assigned the top-scoring module to each cell[16]. This showed that most of this patient's tumour cells were 'Stress (p53 response)'-like, followed by 'Mitochondrial', 'Melanocytic' and 'Antigen Presenting' (Supplementary Fig. 3a, b). However, when we examined the expression of the module-defining genes in our GeoMx dataset by performing unsupervised hierarchical clustering of the ROIs, we observed greater heterogeneity compared to the snRNA-Seq data including increased expression of mesenchymal module genes in ROIs B09-B12 or hypoxia response in ROIs A05-A07 and B01-B02 (Fig. 5e). This discrepancy may arise from the fact that GeoMx captures transcripts from the entire cell, whereas snRNA-Seq reflects nuclear RNA alone, or simply from the use of different parts of the same tumour. Nonetheless, these differences highlight the value of integrating multiple analytical approaches in tumour profiling to achieve a more comprehensive understanding of the tumour's behaviour.

**Immune cell landscape of the primary tumour**

Next, we wished to investigate the immune cell composition of the primary tumour. To do this, we subset the immune cells in our snRNA-Seq dataset and, using the *SingleR* R package, annotated macrophages, T cells, NK cells and dendritic cells (Supplementary Fig. 4a). Out of those, the macrophage population was the most abundant. We then decided to

delineate the M1 and M2 macrophage populations, with the expectation that this tumour would contain more M1 (pro-inflammatory) macrophages, because of the high expression of *MIF* (Fig. 3c). However, we did not see a significant difference in expression of M1 or M2 markers (Supplementary Fig. 4b, c). This could suggest that indeed this tumour contained both tumour-promoting and proinflammatory macrophages, or could be due to the fact that tumour-infiltrating macrophages in melanoma display other marker expression than their counterparts in other tumour entities

To address the heterogeneity of the very scarce T cell population, we utilised the *ProjecTILs* R package to project the T cells onto a well-annotated T cell reference atlas, which enables their direct comparison in a robust manner[17]. This revealed that most cells resemble CD8 + T cells and Th1 cells (Supplementary Fig. 4d, e). Finally, we analysed the expression of immune checkpoint genes, as increased expression could result in immune checkpoint inhibition escape. In the snRNA-Seq data, unsurprisingly, we saw very low levels in the tumour cell clusters 0–4 and 9, but strong enrichment of some genes in the immune cell clusters 5 and 7 (Supplementary Fig. 4f, g). The most enriched gene was *CD86*, known to be a marker of proinflammatory M1 macrophages and dendritic cells, again pointing towards the pro-inflammatory macrophages playing a role in the development of this tumour[35]. Finally, we analysed the expression of those genes in our GeoMx data, which revealed decreased levels of almost all genes, bar *PVR, VSIR* and *NECTIN2*, in the primary tumour, compared to healthy tissues.

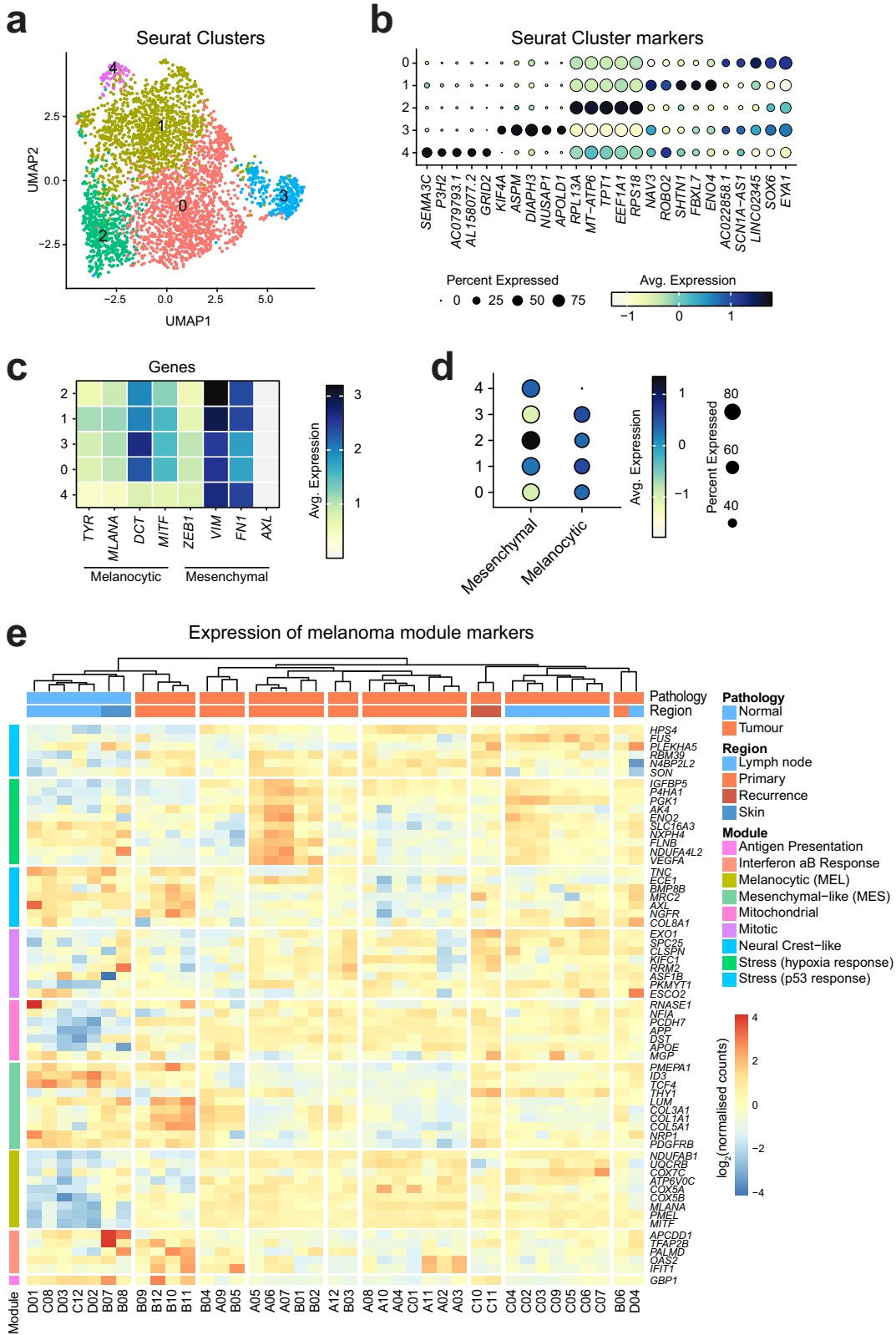

**Fig. 5 | Transcriptional heterogeneity of the primary tumour. a** UMAP visualisation of all malignant cells (*n* = 3659) within the primary tumour, coloured by clusters identified with *Seurat*. **b** Dot plot of normalised expression gene markers of the identified tumour clusters. **c** Heatmap representation of the normalised expression of melanocytic (*TYR, MLANA, DCT* and *MITF*), and mesenchymal (*ZEB1, VIM, FN1, AXL*) genes in the tumour clusters. **d** Dot plot of the grouped expression of the mesenchymal and melanocytic genes from panel c in the tumour clusters. **e** Heatmap representation of non-hierarchical clustering of GeoMx ROIs based on log$_2$(normalised counts) expression of melanoma-module marker genes[16].

## Discussion

While the genetic and treatment landscape in adult melanoma, representing a substantial burden in oncology, has been studied extensively, little is known on the genetic basis of paediatric melanoma[1]. A study from Lu et al. has investigated 23 paediatric melanoma cases by whole exome and whole genome sequencing, unravelling *BRAF* and *NRAS* mutation frequencies of 56% and 11% respectively[23]. These numbers that are comparable to adult disease may represent the genetic basis of the disease. However, other influencing factors which have been implicated in large hallmark studies of adult melanoma (such as *MITF* and *LCK* expression) and which may modify disease course have rarely been described in paediatric cases[4]. The amplification of *MITF* reported in this study highlights a mechanism of oncogene activation which has hitherto not been explored in the paediatric context: Although the resulting *MITF* expression did not appear to be spatially different within the primary tumour, testing for *MITF* amplification in the context of paediatric melanoma is certainly a conclusion from this study. While patients with an MITF-activated transcriptome displayed an intermediate disease course in the TCGA analysis[4], the amplification of *MITF* has been linked to a more severe disease course and metastatic diseases already years ago[19]. Although the cited study dates before the advent of immune checkpoint inhibition, other publications have pointed out the immunosuppressive properties of MITF, reviewed recently in Lee et al.[36].

Although the expression levels of *MITF* and *BRAF* across the analysed ROIs—spanning primary tumours, lymph node metastases, and cutaneous metastases—did not show significant differences, the analysis of a gene signature associated with ICI failure[29] revealed a notable trend. Specifically, most regions of the primary tumour exhibited an upregulation of genes positively linked to resistance (e.g., *PSMB5* and *MSSB2*) and a downregulation of genes negatively associated with resistance. Additionally, expression of *VEGFA*, a known angiogenesis factor, was also spatially variable, with the highest levels in the tumour periphery, potentially causing parts of the primary tumour to be more resistant to immune therapy.

Albeit being a single case study, the spatio-temporal analysis is instructive in several ways: Unsupervised transcriptional analysis comparing the most variable genes throughout all ROIs shows that the melanoma region from the primary tumour, which is most similar to the relapse, localises to the tumour periphery (ROI 'B06'). This substantiates the need for a wider resection of the tumour also in paediatric patients. In fact, the tumour/microenvironment interface of melanoma has recently been shown to be transcriptionally distinct from the tumour bulk[8].

The deep molecular work-up of this case provides both temporal but also spatially a role of the *MITF* amplification (together with the more common *NRAS* mutation and *BRAF* amplification) in initiating the disease. Transcriptional factors (such as ICI-resistance genes) or the presence of specific types of immune cells may have contributed to the resistance to this therapy element. Additionally, WNT pathway upregulation, consistently observed in lymph node metastasis and disease recurrence, or dedifferentiation mechanisms might drive the relapsing tumour towards further aggressiveness and may have caused the fatal disease course.

Overall, our study stimulates further questions which warrant investigation in larger studies of paediatric melanoma, including relapse cases. These pertain to the WNT pathway and VEGFA signalling as potential drivers of metastatic disease, and possibilities to interfere with those pathways. As double-check-point inhibition in this case was not successful, other approaches could involve epigenetically modifying drugs, such as the Pan-HDAC inhibitor Entinostat, in combination with ICI in maintenance treatment.

Albeit our study is limited to a single case and thus cannot support broad therapy-related conclusions, these findings suggest that alternative therapeutic strategies, such as tyrosine kinase inhibitors (TKIs) in combination with other therapies, including ICI, might offer more effective options for patients with tumours presenting this signature. Importantly, our study adds to the very limited information available on paediatric melanoma, including datasets such as spatial transcriptomics, snRNA-Seq and DNA methylation profiles of matched tissues.

In summary, larger studies, including those focused on the tumour microenvironment, are necessary to pave the way for molecularly guided therapy in these refractory melanoma cases into clinical practice.

## Data availability

All raw and processed sequencing and array data have been deposited to the GEO database under accession numbers GSE286410 (sequencing) and GSE286412 (methylation array). A list of significantly differentially expressed genes from Fig. 3a, b is included in the Supplementary Data 1 file.

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

## Acknowledgements

First and foremost, we thank the patient and her family. We also thank the Biobank Facility at the University Hospital Augsburg for their cooperation in sample collection and processing. P.D.J. received funding from Else-Kröner-Fresenius-Stiftung and Max-Eder Programme of the Deutsche Krebshilfe. V.E.F. is funded by the Konrad Adenauer Foundation. M.L. is funded by the Kind-Philipp Foundation.

## Author contributions

M.M., M.C.F. and P.D.J. conceived the study. M.M. and F.D. performed the GeoMx DSP run and library preparation. N.G.R. and B.M. helped with FFPE slide preparation for GeoMx DSP. F.D. and D.L. performed the DNA methylation array and snRNA-Seq. S.D. performed the Sanger sequencing. S.B., K.G., M.C.F. and P.D.J. helped with the patient's treatment history. M.M. performed all bioinformatic analyses, except for DNA methylation CNV analysis, which was performed by M.L. with input from V.E.F. C.S., D.T.S. and I.B.B. provided molecular and clinical counselling on melanoma. T.S. performed the surgery. M.M. and P.D.J. wrote the manuscript with input from all other authors.

## Funding

## Competing interests

N.G.R. received compensation for travel expenses from Nanostring (Bruker Spatial Biology). All other authors declare no competing interests.
