## [Transparent Peer Review file · Communications Medicine]

Multi-modal Omics Analysis of a Paediatric Melanoma Highlights Mechanisms Underlying Treatment Resistance.

Corresponding Author: Professor Pascal Johann

Version 0:

Reviewer comments:

Reviewer #1

(Remarks to the Author)
COMMSMED-25-0268-T

Title: Multi-modal Omics Analysis of a Paediatric Melanoma Highlights Mechanisms Underlying Treatment Resistance

Mucha and colleagues deployed multimodal molecular analyses to highlight potential therapy resistance mechanisms in a rare pediatric melanoma patient. The patient was born with several nevi on the scalp and diagnosed at 16 months of age with congenital malignant melanoma. After lymph node involvement, the lesions were surgically removed and different lines of treatments followed: immune-checkpoint blockade (anti-PD1), targeted therapy (MEKi and mebendazole), and chemotherapy (carboplatin and VP16). Unfortunately, the disease progressed and 10 months after diagnosis, distant metastases were detected in the lungs and spine, before the patient succumbed to the disease.

To identify underlying molecular and cellular mechanisms the authors deployed an impressive combination of state-of-the-art methods, such as Sanger-sequencing, DNA methylation array analyses, snRNA sequencing, and spatial transcriptomics (GeoMx DSP). On the DNA level they revealed a NRASQ61H mutation as well as a larger chromosomal gain (spreading over 5 cytobands) of Chr. 3p, including the MITF locus, in the primary tumor as well as the lymph node metastasis. Spatial and single-nuclei-based transcriptomic analyses led to the identification of WNT-genes upregulated in metastatic and recurrent vs. primary tumor tissue, a VEGF high-expressing region in the primary tumor, and a proliferative subcluster of malignant cells.

Overall, this is a well-presented, designed and executed case study. Mucha and colleagues were able to identify molecular aberrations that potentially drive the aggressive behaviour (resistant phenotype). The results advance the field of generally understudied pediatric melanomas.

In my opinion, the manuscript could benefit from a deeper exploration and integration of the snRNAseq and spatial transcriptomic data.

+ As the ROIs contain potentially different cell types, it would be informative to deconvolute these regions for major cell types (see: <https://www.nature.com/articles/s41467-022-28020-5>). CD163 and ITGAX expression hints towards inflammatory macrophages. Could you plot the 6-gene signature in your snRNA data per cell type? CD163 expression seems also elevated in S100B positive B11 ROI (primary tumor).

+ Besides macrophages, could you please subset the T cell population of the snRNAseq data and check for effector, naïve and exhausted markers? How about the general expression of immune checkpoint molecules in GeoMx and snRNA data of the primary tumor?

+ The Chr.3 amplification is intriguing, especially because of the MITF locus. How is MITF expression and genes of this DNA gain distributed across the different ROIs and snRNAseq data? Did you try to use DNA-FISH to orthogonally validate this finding? If not, it would be informative to infer CNV aberrations from the snRNAseq data (using Copykat or Honeybadger) to cross-validate the Chr.3p gain. These inferred CNVs could then also be plotted spatially across the ROIs for example to trace some kind of clonal evolution.

+ The spatially confined expression of VEGFA is interesting. Do these A5,6,7 VEGFA-high expressing ROIs also express PECAM1? Could you localize blood vessels (endothelial cells) in these sections/ROIs? Basically, are sure that the VEGFA expression signal comes exclusively from melanoma cells?

+ snRNAseq experiment: I can't really follow why the authors concentrated on malignant cluster 4, which describes generally proliferating/dividing melanoma cells. What is the phenotypic heterogeneity of the melanoma cell clusters? (<https://doi.org/10.1016/j.cell.2023.11.037>)

Could you please subset only the malignant Seurat_clusters and recluster them? How many new clusters would you identify? Heatmap of most characteristic markers? Do you see signs of melanoma dedifferentiation? Also here, could try to map different melanoma cell states back to your ROIs?

Reviewer #2

(Remarks to the Author)

The cornerstone of Mucha et al. lies in the comprehensive collection of single-nuclei and spatial multi-omics data on a rare case of ICI resistant pediatric melanoma. In general, the authors have analyzed each data modality, spanning from inferring copy numbers, driver mutations (NRAS/BRAF) and high resolution spatial transcriptome (GeoMx), and elaborated de novo NRAS mutation, MITF amplification, WNT pathway upregulation, VEGFA up-regulations as the key features of the potential causes to the ICI resistance. Overall, this is an interesting approach to personalized medicine at the face of such rare disease such as pediatric melanoma. I suggest the following revisions to further enrich the study:

1. Lack of method descriptions: Detailed descriptions of DESeq2 analysis comparing healthy lymph nodes, primary tumor and metastatic tumors.

2. Lack of dissecting immune cell types in the spatial data sets: While I understand that the authors have focused on analyzing the tumor-intrinsic factors that may have driven the ICI resistance, it still necessitates to characterize at least lymphoid subsets in the tumor microenvironment in the spatial data, hence understand the tumor-extrinsic factors in tandem to provide more complete view of the resistance.

3. Chemokine/cytokine differential expressions: Following from the comments 1 & 2, it necessitates to further elaborate what chemokine/cytokine may have contributed to the resistance. For example, Figure 3D clearly highlights CXCL9 in recurrent tumors, which is known to attract immune cells and promote PD-1/PD-L1 blockade efficacy.

Version 1:

Reviewer comments:

Reviewer #1

(Remarks to the Author)

Mucha and colleagues have substantially improved their manuscript by additional analyses. They addressed all my points adequately. Hence, I would like to recommend this manuscript for publication.

Reviewer #2

(Remarks to the Author)

The authors have addressed the comments succinctly. I have no more concerns.

We thank both Reviewers for their positive evaluation of our work and for the useful suggestions. We have now significantly revised the manuscript to include the new analysis and figures. A point-by-point response to the questions can be found below, and all changes to the manuscript have been highlighted in blue in the new document.

Reviewers' comments:

Reviewer #1 (Remarks to the Author):

COMMSMED-25-0268-T

Title: Multi-modal Omics Analysis of a Paediatric Melanoma Highlights Mechanisms Underlying Treatment Resistance

Mucha and colleagues deployed multimodal molecular analyses to highlight potential therapy resistance mechanisms in a rare pediatric melanoma patient. The patient was born with several nevi on the scalp and diagnosed at 16 months of age with congenital malignant melanoma. After lymph node involvement, the lesions were surgically removed and different lines of treatments followed: immune-checkpoint blockade (anti-PD1), targeted therapy (MEKi and mebendazole), and chemotherapy (carboplatin and VP16). Unfortunately, the disease progressed and 10 months after diagnosis, distant metastases were detected in the lungs and spine, before the patient succumbed to the disease.

To identify underlying molecular and cellular mechanisms the authors deployed an impressive combination of state-of-the-art methods, such as Sanger-sequencing, DNA methylation array analyses, snRNA sequencing, and spatial transcriptomics (GeoMx DSP). On the DNA level they revealed a NRASQ61H mutation as well as a larger chromosomal gain (spreading over 5 cytobands) of Chr. 3p, including the MITF locus, in the primary tumor as well as the lymph node metastasis. Spatial and single-nuclei-based transcriptomic analyses led to the identification of WNT-genes upregulated in metastatic

and recurrent vs. primary tumor tissue, a VEGF high-expressing region in the primary tumor, and a proliferative subcluster of malignant cells.

Overall, this is a well-presented, designed and executed case study. Mucha and colleagues were able to identify molecular aberrations that potentially drive the aggressive behaviour (resistant phenotype). The results advance the field of generally understudied pediatric melanomas. In my opinion, the manuscript could benefit from a deeper exploration and integration of the snRNAseq and spatial transcriptomic data.

1. As the ROIs contain potentially different cell types, it would be informative to deconvolute these regions for major cell types (see: <https://www.nature.com/articles/s41467-022-28020-5>). CD163 and ITGAX expression hints towards inflammatory macrophages. Could you plot the 6-gene signature in your snRNA data per cell type? CD163 expression seems also elevated in S100B positive B11 ROI (primary tumor).

We thank the Reviewer for these suggestions. To address the first point, we performed SpatialDecon of the GeoMx ROIs based on the cell types annotated in our snRNA-Seq from the same patient. This confirmed our annotation of ROIs B07, B08, C08, C12, D01-D03 as not containing any tumour cells, and instead composed primarily of microenvironment cells. Interestingly, ROI B12 was also predicted to not contain any tumour cells, despite mapping to the primary tumour sample. Additionally, this ROI showed the closest similarity to the healthy ROIs in our UMAP projection in Fig. 3b (Revised Supplementary Fig. 1b), and clustered closely with healthy tissues in the expression of the 6 ICI-resistance implicated genes in Fig. 4a (Revised Fig. 3d).

Rebuttal Fig. 1. Spatial Deconvolution of GeoMx ROIs into cell types identified through matched snRNA-Seq, performed with the SpatialDecon R package¹.

Surprisingly though, the tumour content in the ROIs located in the primary, recurrence and metastasis samples was estimated to be relatively low, ~30-40% in most, despite being strongly stained by the S100/Pmel17 antibody (Fig. 3a, also showed below). Additionally, ROIs which showed the strongest staining with a CD45 antibody in the primary tumour (e.g. B02, B03) were not predicted by SpatialDecon to contain a higher number of blood cells compared to other ROIs.

Spatial distribution of Regions of Interest (ROIs) profiled in this study

Original Fig. 3a (Revised Supplementary Fig. 1a). Spatial distribution of Regions of Interest (ROIs) profiled in this study.

Therefore, while we agree that cell type deconvolution of the ROIs would be beneficial, we believe that in order for it to be more accurate, one would have to perform an additional stain for, for example, endothelial cells using a PECAM1 antibody, and use segmentation of ROIs into S100+/CD45-/PECAM1-, CD45+/S100-, or PECAM1+/S100-/CD45- cells, which we unfortunately did not perform in the initial experiment. However, the accompanying snRNA-Seq of the primary tumour provides insights into the transcriptional programs stemming specifically from the tumour or microenvironment cells.

To address the second point regarding expression of the 6 ICI-resistance implicated genes in the snRNA-Seq dataset, the following dot plot confirms that tumour cells express higher levels of the *PSMB5* and *MTSS2* genes, and conversely lower levels of *TNFAIP2*, *ITGAX*, *SAMSN1* and *CD163*, compared to microenvironment. Specifically, as expected, the latter genes are most expressed in the macrophage cell population, suggesting indeed the enhanced presence of inflammatory macrophages in the tumour mass.

Revised Fig. 3e. Dot plot of scaled snRNA-Seq expression levels of *TNFAIP2*, *ITGAX*, *SAMSN1*, *CD163*, *PSMB5* and *MTSS2* in the indicated cell types.

2. Besides macrophages, could you please subset the T cell population of the snRNAseq data and check for effector, naïve and exhausted markers? How about the

general expression of immune checkpoint molecules in GeoMx and snRNA data of the primary tumor?

We thank the Reviewer for these suggestions and agree that a more thorough analysis of the immune cell landscape of the tumour would be important. As suggested, we now subset the immune cell clusters in our snRNA-Seq dataset and using the *SingleR* R package annotated Macrophages, T cells, NK cells and Dendritic cells (New Supplementary Fig. 4a). Out of those, the macrophage population was the largest. We then decided to delineate the M1 and M2 macrophage populations, with the expectation that this tumour would contain more M1 (pro-inflammatory) macrophages, as noted in the previous question. However, we did not see a significant difference in expression of M1 or M2 markers in the macrophage population (New Supplementary Fig. 4b, c). This could suggest that indeed this tumour contained both tumour-promoting and proinflammatory macrophages. A further in-depth exploration in a larger number of melanoma cases would be necessary to resolve this issue, given the still relatively limited number of cells in this case.

Supplementary Fig. 4a, b and c.

To address the heterogeneity of the very scarce T cell population, we utilised the *ProjectTILs* R package to project the T cells onto a well-annotated T cell reference atlas, which enables their direct comparison². This revealed that most cells resemble CD8+ T cells (cytotoxic) and Th1 cells (New Supplementary Fig. 4d, e).

Supplementary Fig. 4d and e. Characterisation of the T cell populations in the snRNA-Seq data.

Regarding the immune checkpoint molecules, we first analysed their expression in the snRNA-Seq data, which unsurprisingly revealed very low expression in the tumour cell clusters 0-4 and 9, but strong enrichment of some genes in the immune cell clusters 5 and 7 (New Supplementary Fig. 4f, g). The most enriched gene was *CD86*, known to be a marker of proinflammatory M1 macrophages and dendritic cells, again pointing towards the pro-inflammatory macrophages playing a role in development of this tumour.

Supplementary Fig. 4f and g. Expression of immune checkpoint genes in the snRNA-Seq data.

Finally, we analysed the expression of those genes in our GeoMx data, which revealed decreased levels of almost all genes, bar *PVR*, *VSIR* and *NECTIN2*, in the primary tumour, compared to healthy tissues.

Supplementary Fig. 4h. Expression of immune checkpoint genes in the GeoMx data.

While the conclusions to be drawn from the immune cell analysis presented here are limited mainly due to the low number of cells, we would like to point out that we are currently planning a project with a more robust analysis involving multiple paediatric melanoma patients. This new project will involve deeper analysis of the microenvironment landscape in melanoma.

3. The Chr.3 amplification is intriguing, especially because of the MITF locus. How is MITF expression and genes of this DNA gain distributed across the different ROIs and snRNAseq data? Did you try to use DNA-FISH to orthogonally validate this finding? If not, it would be informative to infer CNV aberrations from the snRNAseq data (using

Copykat or Honeybadger) to cross-validate the Chr.3p gain. These inferred CNVs could then also be plotted spatially across the ROIs for example to trace some kind of clonal evolution.

We thank the Reviewer for these useful suggestions. We now plotted the expression of *MITF* in the GeoMx ROIs as a bubble plot, which confirmed higher expression in the tumour tissues (see below). Moreover, we agree that cross-validating the CNV using a different method would be beneficial and have therefore performed CNV analysis based on the snRNA-Seq data, using the *InferCNV* R package. This confirmed the partial Chr.3p gain also on the transcription level, albeit the visualisation was slightly obscured by concurrent adjacent losses on the same Chr.3p arm (visible as well on the original Fig. panel 2c (Revised Fig. 2d):

Original Fig. 2c, Revised Fig. 2d.
Local amplification of chromosome 3p and concurrent deletions of *MITF*-surrounding cytobands.

Revised Fig. 2f. snRNA-Seq based CNV analysis.

Additionally, while performing this analysis and looking back at the DNA methylation-inferred CNVs, we discovered that Chr.7q was also amplified, amongst others. This is intriguing because Chr.7q includes *BRAF*, a known melanoma oncogene (seen above in the snRNA-Seq based CNV analysis, as well as below in the DNA methylation-based analysis). While the full length amplification of chr.7 is not specific, the involvement of *BRAF* may well represent a further resistance factor as described above.

Revised Fig. 2d. Amplification of chr.7.

The amplification could likely be the reason for resistance to the MEK inhibitor Trametinib, which the patient received in combination with Mebendazole^{3,4}.

In view of this important finding, we have now revised the manuscript and the relevant Fig. to include the *BRAF* amplification and its potential implication in the response to MEK inhibition.

We then checked expression of *MITF* and *BRAF* in our snRNA-Seq and GeoMx datasets, which confirmed increased levels of those transcripts in the tumour tissues:

Revised Fig. 2g. Dot plot of expression levels of *MITF*, *BRAF* and *NRAS* in the snRNA-Seq cell types.

Supplementary Fig. 1c. Related to Fig. 2. Expression of *MITF* and *BRAF* in the GeoMx spatial dataset.

4. The spatially confined expression of *VEGFA* is interesting. Do these A5,6,7 *VEGFA*-high expressing ROIs also express *PECAM1*? Could you localize blood vessels (endothelial cells) in these sections/ROIs? Basically, are sure that the *VEGA* expression signal comes exclusively from melanoma cells?

We thank the Reviewer for these relevant questions. We agree that confirming that the increased *VEGFA* expression comes from true melanoma cells and not TME is important and have now checked the high-resolution images obtained with GeoMx DSP for presence of blood vessels. While we did not stain for *PECAM1*, we can indirectly see that out of the ROIs with the highest *VEGFA* expression (B02 and A05-A07) ROIs A05-A07 contain very little CD45 staining, while ROI B02 contained high number of immune cells (as indicated in panel b below by the CD45 staining). Additionally, in those ROIs we do not see any blood vessel structures. Conversely, expression of *PECAM1* was the highest in the highly infiltrated ROIs belonging to healthy tissues (C08, C12, D01-D03). We confirmed these findings in the snRNA-Seq dataset, where, as expected, *PECAM1* expression levels were the highest in the endothelial cell population, followed by macrophages, whereas *VEGFA* expression levels were highest in the macrophages and tumour cells (panels c and d below).

Therefore, we believe that the *VEGFA* expression comes from melanoma cells. These results are now included in the manuscript in Supplementary Fig. 2.

Supplementary Fig. 2. *VEGFA* expression comes from melanoma cells.

5. snRNAseq experiment: I can't really follow why the authors concentrated on malignant cluster 4, which describes generally proliferating/dividing melanoma cells. What is the phenotypic heterogeneity of the melanoma cell clusters? (<https://doi.org/10.1016/j.cell.2023.11.037>)

Could you please subset only the malignant Seurat_clusters and recluster them? How many new clusters would you identify? Heatmap of most characteristic markers? Do you see signs of melanoma dedifferentiation? Also here, could try to map different melanoma cell states back to your ROIs?

We thank the Reviewer for the useful suggestions and have performed a more detailed analysis of the tumour cell subclusters, which now replaced the former Fig. 6 (Fig. 5 and Supplementary Fig. 3 in the updated manuscript). Specifically, we subset only the malignant cells and reclustered using resolution = 0.3. This identified 5 subclusters, characterized by expression of markers such as *EYA1* (tumour cluster 0), *ENO4* (tumour cluster 1), *TPT1* (tumour cluster 2), *DIAPH3* (tumour cluster 3), and *GRID2* (tumour cluster 4). Expression of these markers suggests highly proliferative, DNA-damage induced

signatures, as expected from a malignant tumour. Importantly, all clusters expressed high levels of mitochondrial and DNA-damage related genes.

Revised Fig. 5a and b. Subclustering of the primary tumour cells in snRNA-Seq.

We then assigned each cell to the modules defined in the suggested publication and assigned the top scoring module to each cell⁵. This showed that the vast majority of this patient’s tumour cells were “Stress (p53 response)”-like, followed by “Mitochondrial”, “Melanocytic” and “Antigen Presenting”.

Supplementary Fig. 3a and b. Related to Fig. 5. Assignment of melanoma modules.

However, when we examined the expression of the module-defining genes in our GeoMx dataset by performing unsupervised hierarchical clustering of the ROIs, we observed greater heterogeneity compared to the snRNA-Seq data including increased expression of mesenchymal module genes in ROIs B09-B12 or hypoxia response in ROIs A05-A07 and B01-B02 (see Revised Fig. 5e below). This discrepancy may arise from the fact that GeoMx captures transcripts from the entire cell, whereas snRNA-Seq reflects nuclear RNA alone, or simply from the use of different parts of the same tumour. Nonetheless, these differences highlight the value of integrating multiple analytical approaches in tumour profiling to achieve a more comprehensive understanding. Importantly, for even more comprehensive analysis we are currently planning to include more patient samples.

Revised Fig. 5e. Expression of melanoma module markers in the GeoMx datasets.

Regarding melanoma de-differentiation, we profiled the expression of melanocytic (i.e. differentiated) and mesenchymal (i.e. de-differentiated) genes in the snRNA-Seq tumour sub-clusters. This revealed potential de-differentiation of cells within tumour cluster 4, which expressed lower levels of melanocytic markers (see below). Therefore, while we agree that this tumour shows some signs of de-differentiation, a larger study would be needed to determine its significance in paediatric melanomas.

Revised Fig. 5c and d. Expression of de-/differentiation markers in the tumour sub-clusters.

Reviewer #2 (Remarks to the Author):

The cornerstone of Mucha et al. lies in the comprehensive collection of single-nuclei and spatial multi-omics data on a rare case of ICI resistant pediatric melanoma. In general, the authors have analyzed each data modality, spanning from inferring copy numbers, driver mutations (NRAS/BRAF) and high resolution spatial transcriptome (GeoMx), and elaborated de novo NRAS mutation, MITF amplification, WNT pathway upregulation, VEGFA up-regulations as the key features of the potential causes to the ICI resistance. Overall, this is an interesting approach to personalized medicine at the face of such rare disease such as pediatric melanoma. I suggest the following revisions to further enrich the study:

1. Lack of method descriptions: Detailed descriptions of DESeq2 analysis comparing healthy lymph nodes, primary tumor and metastatic tumors.

We apologize for this oversight and have now included detailed description of the analysis performed with DESeq2 in the revised methods section of the manuscript. The updated text reads as follows: “Differential gene expression analysis between Metastasis/Recurrence and Primary tumour ROIs was performed on the Q3-normalised data with *DESeq2* v.1.42.1, with significantly deregulated genes defined by $\text{abs}(\log_2\text{FoldChange}) > 1$ & $\text{baseMean} > 15$ & $\text{padj} < 0.0530$. Results were visualized using the *EnhancedVolcano* and *pheatmap* R packages.”

2. Lack of dissecting immune cell types in the spatial data sets: While I understand that the authors have focused on analyzing the tumor-intrinsic factors that may have driven the ICI resistance, it still necessitates to characterize at least lymphoid subsets in the tumor microenvironment in the spatial data, hence understand the tumor-extrinsic factors in tandem to provide more complete view of the resistance.

We thank the reviewer for this insightful comment. We fully agree that a comprehensive understanding of ICI resistance would benefit from jointly characterizing tumor-intrinsic and tumor-extrinsic factors, including detailed profiling of immune cell subsets in the spatial context. Unfortunately, in the current study, the spatial data do not allow for a reliable and granular dissection of immune cell types due to our initial focus on the tumour cells only - and the selection of the ROIs in accordance with this priority. However, we recognize the importance of this analysis and are currently planning follow-up studies with expanded patient cohorts and enhanced spatial profiling methods to specifically address this limitation.

In the current revised manuscript, we instead provide an in-depth single-nucleus RNA-seq analysis of the primary tumour, which offers high-resolution insights into immune cell populations, including lymphoid subsets (Supplementary Fig. 4). We have now

clarified this in the revised text and we hope this helps strengthen the tumour microenvironment component of the study.

3. Chemokine/cytokine differential expressions: Following from the comments 1 & 2, it necessitates to further elaborate what chemokine/cytokine may have contributed to the resistance. For example, Fig. 3D clearly highlights CXCL9 in recurrent tumors, which is known to attract immune cells and promote PD-1/PD-L1 blockade efficacy.

We thank the Reviewer for the suggestion to check the expression of cytokine and chemokine-encoding genes, which could be highly indicative of therapy resistance mechanisms. Below, we plotted the expression of all those genes, covered by the Nanostring WTA panel and which were not filtered out during the QC steps. However, as we can see in the heatmap, the only genes showing higher expression in the tumour tissues (primary, metastasis or recurrence) are *VEGFA*, as noted previously, *GDF11* and *MIF*. The potential influence of the spatially heterogeneous expression of *VEGFA* was already discussed in our manuscript. However, *MIF* expression pattern is also interesting, as it is highly upregulated in affected tissues. High expression of *MIF* (macrophage migration inducible factor) in melanoma cells has previously been linked to increased chance of metastasis⁶, which suggests it to be another potential reason for a very aggressive course of the disease in this patient, adding to the already complicated picture. We have now included this panel in Revised Fig. 3c, and made changes in the text accordingly.

Revised Fig. 3c. Expression of cytokine-encoding genes in the GeoMx ROIs.

Rebuttal letter references:

1. Danaher, P. *et al.* Advances in mixed cell deconvolution enable quantification of cell types in spatial transcriptomic data. *Nat. Commun.* 2022 131 **13**, 1–13 (2022).
2. Andreatta, M. *et al.* Interpretation of T cell states from single-cell transcriptomics data using reference atlases. *Nat. Commun.* 2021 121 **12**, 1–19 (2021).
3. Manzano, J. L. *et al.* Resistant mechanisms to BRAF inhibitors in melanoma. *Ann. Transl. Med.* **4**, 1–9 (2016).
4. Stagni, C. *et al.* BRAF gene copy number and mutant allele frequency correlate with time to progression in metastatic melanoma patients treated with MAPK inhibitors. *Mol. Cancer Ther.* **17**, 1332–1340 (2018).
5. Pozniak, J. *et al.* A TCF4-dependent gene regulatory network confers resistance to immunotherapy in melanoma. *Cell* **187**, 166-183.e25 (2024).
6. Soumoy, L., Kindt, N., Ghanem, G., Saussez, S. & Journe, F. Role of Macrophage Migration Inhibitory Factor (MIF) in Melanoma. *Cancers (Basel)*. **11**, 529 (2019).